# Differential Gene and Protein Expressions Responsible for Vasomotor Signaling Provide Mechanistic Bases for the Opposite Flow-Induced Responses of Pre- and Post-Circle of Willis Arteries

**DOI:** 10.3390/life15060856

**Published:** 2025-05-26

**Authors:** Zoltan Nemeth, Krisztian Eros, Gyongyi Munkacsy, Akos Koller

**Affiliations:** 1HUN-REN-SE, Cerebrovascular and Neurocognitive Disease Research Group, Institute of Translational Medicine, Semmelweis University, 1085 Budapest, Hungary; znemeth@umc.edu; 2Department of Biochemistry and Medical Chemistry, Medical School, University of Pecs, 7624 Pecs, Hungary; krisztian.eros@aok.pte.hu; 3Szentagothai Research Centre, University of Pecs, 7624 Pecs, Hungary; 4Department of Bioinformatics, Semmelweis University, 1085 Budapest, Hungary; munkacsy.gyongyi@med.semmelweis-univ.hu; 5Institute of Molecular Life Sciences, HUN-REN Research Centre for Natural Sciences, 1117 Budapest, Hungary; 6HUN-REN-SE, Cerebrovascular and Neurocognitive Disease Research Group, Department of Morphology and Physiology, Institute of Translational Medicine, Semmelweis University, 1085 Budapest, Hungary; 7Research Center for Sport Physiology, Hungarian University of Sports Science, 1123 Budapest, Hungary; 8Department of Physiology, New York Medical College, Valhalla, NY 10595, USA

**Keywords:** cerebral blood flow regulation, mechano-transduction, small cerebral arteries, arachidonic acid metabolism, nitric oxide signaling

## Abstract

Increases in flow elicit dilations in the basilar artery (BA) supplied by the posterior cerebral circulation (PCC), and ensuring efficient blood supply to the circle of Willis in which blood flow and pressure can distribute and equalize, and thus provide the appropriate supply for the daughter branches to reach certain brain areas. In contrast, increases in flow elicit constrictions in the middle cerebral artery (MCA), supplied by the anterior cerebral circulation (ACC) and regulating the blood pressure and flow in distal cerebral circulation. Mediators of flow-dependent responses include arachidonic acid (AA) metabolites and nitric oxide (NO). We hypothesized that mediators of flow-dependent responses are differentially expressed in cerebral arteries of the PCC (CA_PCC_) and ACC (CA_ACC_). The expressions of key enzymes of the AA pathway—cyclooxygenases (COX1/COX2), cytochrome P450 hydroxylases (Cyp450), thromboxane synthase (TXAS), thromboxane A2 (TP) receptor, prostacyclin synthase (PGIS), prostacyclin (IP) receptor (IP); neuronal nitric oxide synthase (nNOS), and endothelial nitric oxide synthase (eNOS)—in the BA and MCA from rats (n = 20) were determined by western blotting. Transcriptome analysis in CA_PCC_ and CA_ACC_ from rats (n = 25) was assessed by RNA sequencing. In BA compared to MCA, COX1/2 and Cyp450 protein expressions were lower, PGIS was higher, TXAS and nNOS/eNOS were similar, TP receptors were lower, and IP receptors were higher. Gene expressions of vasodilator canonical pathways were higher in CA_PCC_; vasoconstriction canonical pathways were higher in CA_ACC_. Mediators of flow-dependent vasomotor signaling are differentially expressed in cerebral arteries of the posterior and anterior circulation, corresponding to their vasomotor function.

## 1. Introduction

Closely regulated cerebral blood flow is essential for maintaining optimal brain function, which is achieved by different and appropriate vasomotor functions of the posterior cerebral circulation (PCC), such as the basilar artery (BA), and in the anterior cerebral circulation (ACC), such as the middle cerebral artery (MCA) [1]. Cerebral arteries of the PCC (CA_PCC_) system provide increased or decreased blood flow to the circle of Willis, in which the pressure and flow equilibrate, ensuring a constant supply of cerebral blood flow to distal branches.

In contrast, the role of cerebral arteries of the ACC (CA_ACC_) is to prevent great increases in intracranial pressure and flow, as the closed cranium limits changes in blood flow/volume, a principle recognized early on by Monro and Kellie [2]. The constancy of total intracranial volume is in part achieved by the autoregulation of cerebral blood flow via pressure- and flow-induced vasomotor function (dilation/constriction), which prevents the transmission of high systemic pressures and flow into the cerebral capillary network [3,4,5,6]. Indeed, a majority of previous studies have shown that increases in flow elicit dilations in BA, whereas constrictions in MCA limit volume changes [4,7,8,9]. It has been observed that different cerebral vessels respond to increases in flow with dilation, constriction, or in a biphasic manner [7,10,11]. For example, the isolated rat BA dilates, whereas the MCA constricts to increases in intraluminal flow [4,12].

Several molecules are implicated in the mediation of flow-induced vasomotor responses of cerebral vessels, including vasoactive arachidonic acid (AA) metabolites [4]. The importance of vasoactive AA metabolites, including the vasoconstrictor 20-hydroxyeicosatetraeonic acid (20-HETE), thromboxane A2 (TXA2), and the vasodilator prostacyclin (PGI2) in mediating the cerebrovascular tone has been widely investigated [13,14,15,16]. For example, we and others have shown that 20-HETE is essential for the normal flow-induced vasoconstriction responses of the isolated human and rat MCA [4,15,16,17,18,19].

A role for nitric oxide (NO) in modulating cerebral autoregulation by causing arterial dilation to compensate for flow-induced vasoconstriction has also been suggested [8]. Interestingly, the role and presence of intracellular enzymes and genes responsible for the production of mediators and/or modulators of flow-dependent responses of cerebral arteries of the posterior and anterior circulation, however, have not yet been revealed.

Based on the above, we hypothesized that enzymes and genes that are responsible for the production of vasomotor mediator molecules involved in flow-dependent responses are different and opposite in cerebral arteries of the posterior, such as the BA and the anterior cerebral circulation, such as the MCA.

Thus, we aimed to determine the protein expression of key enzymes and receptors by using Western blot and gene expression using RNA-sequencing in cerebral arteries of the posterior and anterior cerebral circulations to see whether they correspond to the opposite flow-induced vasomotor functions.

## 2. Materials and Methods

### 2.1. Animals

A total of 45 Wistar–Kyoto rats (male, 3 months old) were purchased from Toxi-Coop Zrt (Budapest, Hungary), delivered to the Animal Facility of Semmelweis University, housed under controlled humidity (30–70%), temperature (25 ± 2 °C), and constant light cycle (12 h light/dark), and allowed free access to a standard rat chow diet and water. The rats underwent an acclimation period of 3–7 days before being used in experiments. At the time of sacrifice, their body weight ranged from 250–350 g, consistent with the previous study by Toth et al., 2011 [4].

### 2.2. Isolation of Cerebral Arteries

The rats were anesthetized by isoflurane inhalation at a concentration of 3% in inspired compressed air using a vaporizer machine, ensuring deep anesthesia, and then decapitated using a rat guillotine. Decapitation was performed in an isolated area to prevent the animals from witnessing each other’s decapitation, ensuring ethical and physiological/behavioral considerations. The rats designated for Western blot and RNA-sequencing analyses were sacrificed at different time points while maintaining consistent anesthetic protocols, vessel harvesting conditions, and consistent time each day to minimize circadian rhythm influences on expression patterns.

The brains were removed and placed in ice-cold PBS (pH~7.4). For Western blot analysis, MCA and BA segments from 20 rats, and for RNA-sequencing analysis, the surface cerebral arteries of the ACC (circle of Willis and its branches and MCA), and BA from 25 rats were dissected with sterilized microsurgical tools, ensuring to remove all surrounding tissues (e.g., meninges and brain tissues). The arteries were washed in cold PBS to remove blood, transferred separately into sterile Eppendorf tubes (arteries of the anterior cerebral circulation and basilar arteries), and then transferred into liquid nitrogen and stored at −80 °C until use.

For the Western blot detection of AA enzymes and receptors, and NOS enzymes in BA and MCA, snap-frozen BA and MCA segments were homogenized in 200 µL of ice-cold radioimmunoprecipitation assay (RIPA) lysis buffer (composed of 50 mM/L Tris-HCl (pH 7.4), 150 mM/L NaCl, 1.0 mM/L EDTA, 1% Triton X-100, 0.1% SDS, 0.25% SDC) and supplemented with protease (2%, 11836145001 cOmplete Protease Inhibitor, Sigma Aldrich, Saint Louis, MO, USA) and phosphatase (1%, P5726, Sigma) inhibitor cocktails. Lysates were sonicated and centrifuged at 20,000 rcf for 20 min at 4 °C. Then, the supernatant was collected, and protein content was determined using the Bradford assay (ThermoFisher Scientific, Waltham, MA, USA). The 5× Laemmli sample buffer was added to the protein sample at 10%, boiled for 10 min at 100 °C, and stored at −20 °C until use.

Protein samples (10 µg) were run on polyacrylamide gels (8–12%) for 1.5 h at 100 V. Separated proteins were transferred to PVDF membranes (88518, ThermoFisher Scientific) for 2 h at 90 V. ProSieve QuadColo protein marker (00193837, Lonza Bioscience, Walkersville, MD, USA) was used to estimate the molecular mass of the bands. The membranes were incubated with blocking buffer (composed of 1X PBS, 0.1% Tween-20, 5% *w*/*v* nonfat dry milk) for 1 h at RT, and then rinsed with T-PBS for 25 min, and then incubated with primary antibodies, including Rabbit anti-COX1 (1:750, ab227513, Abcam, Cambridge, UK), rabbit anti-COX2 (1:750, ab15191, Abcam), rabbit Anti-Cytochrome P450 4A/CYP4A11 (1:100, ab3573, Abcam), rabbit anti-thromboxane synthase (TXAS, 1:200, ab39362, Abcam Rabbit Anti-Thromboxane A2 receptor (TBXA2R, 1:500, ab233288, Abcam), rabbit anti-PGIS (1:2000, ab23668, Abam), rabbit anti-PTGIR (1:200, PA5-120645, Thermo Fisher Scientific), rabbit anti-nNOS (1:500, 4234S, Cell Signaling, Danvers, MA, USA), and rabbit anti-eNOS (1:100, ab5589, Abcam), overnight at 4 °C. Mouse anti-β-actin (1:20,000, ab6276, Abcam, 1 h at RT) and mouse anti-vinculin (1:1000, 14-9777-82, Thermo Fisher Scientific, overnight at 4 °C) were used as loading controls. The membranes were rinsed with T-PBS for 25 min, and then incubated with HRP-conjugated horse anti-mouse (1:60,000, 7076S, Cell Signaling), and goat anti-rabbit (1:1000-2500, 7074S, Cell Signaling) IgG for 1 h at RT, and then rinsed with T-PBS for 25 min and incubated with chemiluminescent peroxidase substrate (WBKLS0500, Immobilon, Sigma) for 5 min at RT. Antibody labeling was visualized using the Azure Biosystems imaging system (Dublin, CA, USA). Signal density was quantified using ImageJ 1.53t software. Protein expression was expressed as the ratio of target protein/β-actin or vinculin density.

### 2.3. RNA Sequencing of Cerebral Arteries of PCC and ACC

#### 2.3.1. RNA Extraction

RNA was isolated from arterial tissue using the RNeasy Micro Kit (Qiagen, Hilden, Germany) according to the manufacturer’s instructions, followed by DNA digestion with DNase I solution (Qiagen, Hilden, Germany) to achieve the best RNA quality during isolation. RNA quantity was measured by Qubit 3.0 fluorometer (ThermoFisher, Waltham, USA) with Qubit RNA BR Assay Kit. The RNA quality check was done using the Tapestation 4150 instrument with RNA ScreenTape Analysis (Agilent, Santa Clara, CA, USA). To ensure suitability for RNA sequencing, only samples with an RNA Integrity Number (RIN) ≥ 6 were considered acceptable for library preparation.

#### 2.3.2. RNA Sequencing Analysis

Total RNA from CA_PCC_ and CA_ACC_ samples was processed with the Illumina Stranded mRNA Prep Kit (Illumina, San Diego, CA, USA) according to the manufacturer’s instructions. mRNA has been enriched using oligo-dT-attached magnetic beads before cDNA synthesis was performed. Then, the fragments were adenylated, and Illumina sequencing adapters were ligated onto them. Each sample was indexed with IDT for Illumina RNA UD Indexes (Illumina, San Diego, USA). Finally, the samples were cleaned and amplified by PCR using the reagents and conditions provided in the Illumina Stranded mRNA Prep protocol. The library quantity was determined using the Qubit 3.0 fluorometer (ThermoFisher, Waltham, USA), and the quality was assessed using the Tapestation 4150 system (Agilent, Santa Clara, USA). Only samples with an RNA Integrity Number (RIN) of ≥6 were included in the analysis. Sequencing was performed in an Illumina NextSeq 500 instrument (Illumina, San Diego, USA) using the NextSeq500/550 High Output v2.5 (150 Cycles) sequencing kit.

#### 2.3.3. Bioinformatic Analysis

Bioinformatic analysis of sequenced reads was performed using the Galaxy platform. Before gene expression analysis, the FASTQ files were examined using the FASTQC tool. The mean quality score of reads was improved by the Cutadapt (Galaxy Version 4.4+galaxy0) software [20]. Reads were aligned to mRatBN7.2/rn7 2020 reference genome using the HISAT2 alignment tool [21], and the reads were counted using featureCounts [22]. DESeq2 was used to identify differentially expressed genes between CA_ACC_ and CAPCC samples, with CA_PCC_ serving as the baseline [23].

On differentially expressed genes [DEGs, fold change (absolute value) > 1.5 and FDR adjusted *p*-value < 0.05], Ingenuity Pathway Analysis (IPA) (Qiagen) was used to identify canonical pathways related to vascular tone regulation that are impacted in CA_ACC_ samples relative to CA_PCC_ samples. The significance values of pathway enrichment were adjusted by the Benjamini–Hochberg method, and *p* < 0.05 was considered to be statistically significant.

### 2.4. Gene Ontology (GO) Analysis

To further classify the differentially expressed genes (DEGs) based on their functions related to vascular tone regulation, GO analysis was performed using the Mouse Genome Informatics (MGI) platform (Available online: https://www.informatics.jax.org/vocab/gene_ontology (accessed on 21 November 2024)). The following GO Terms were applied: positive regulation of vasoconstriction (GO:0045907), vasodilation (GO:0042311), negative regulation of vasoconstriction (GO:0045906).

### 2.5. Statistical Analysis and Calculations

Statistical analysis was performed using GraphPad Prism v9 (GraphPad, Boston, MA, USA). Normality of the distribution was determined using the Shapiro–Wilk test. Data are expressed as mean ± SEM. Protein expression experiments were reproduced from three individual samples. Each individual sample was taken from six animals, and each sample was run in one to two separate trials. Representative blots and group data are shown. Data were analyzed by a parametric two-tailed independent *t*-test. A value of *p* < 0.05 was considered statistically significant.

The Wald test was used to show differences in gene expression in CA_ACC_ samples relative to CA_PCC_ samples, where CA_PCC_ serves as the baseline for all genes. A value of *p* < 0.05 was considered significant. Gene expression difference was expressed as minus (−) and plus (+) fold change (FC) [log2] signs where −(FC) [log2] represents lower expression of genes in CA_ACC_ relative to CA_PCC_, and +(FC) [log2] represents higher expression of genes in CA_ACC_ relative to CA_PCC_ samples. To examine the functional power of differentially expressed genes, the Wallenius method integrated into goseq was used.

## 3. Results

### 3.1. Protein Levels of Enzymes and Receptors of AA Cascade in BA and MCA

#### 3.1.1. Vasoconstrictor Enzymes and Receptors

Expression of COX1 and COX2 is lower in BA than in MCA arteries

To evaluate if enzymes responsible for producing prostanoid thromboxane and eicosanoids are expressed differentially in BA and MCA arteries, we first examined the protein levels of COX1 and COX2 isoenzymes by Western blotting (Figure 1A,B). Western blot analysis detected significantly lower bands at approximately 70 kDa in BA compared to MCA arteries, using both anti-COX1 and anti-COX2 antibodies, indicating lower expression levels of COX1 and COX2 in BA than in MCA (Figure 1A,B).

Protein level of thromboxane A2 synthase is similar, whereas the thromboxane A2 receptor is lower in BA than in MCA arteries

To determine if downstream vasoconstrictor products of the COX pathway, the constrictor TXA2 production is different in BA and MCA arteries, we measured the protein levels of TXAS and thromboxane A2 receptor (TBXA2R).

We found that the TXAS level was similar between BA and MCA (Figure 1C), while the TP receptor level was significantly lower in BA than MCA (Figure 1D).

The protein level of Cyp4A is lower in BA than in MCA arteries

To examine a potential role for the vasoconstrictor 20-HETE in the flow-induced vasomotor responses of BA and MCA, we measured the protein level of CYP450 4A, known to produce 20-HETE in the rat cerebral vasculature [19]. Western blot analysis detected a significantly lower band at approximately 85 kDa in BA than MCA, indicating lower levels of CYP450 4A protein in BA compared to MCA (Figure 1E).

#### 3.1.2. Vasodilator Enzymes and Receptors

Protein levels of prostacyclin synthase and prostacyclin receptor are higher in BA than in MCA

To further examine the downstream products of the COX pathway, focusing on the vasodilator enzymes and receptors, we determined the protein levels of prostacyclin synthase (PGIS) and prostacyclin receptor (PTGIR) in BA and MCA (Figure 2A,B). Western blot analysis detected bands for PGIS at approximately 56 kDa and PTGIR at approximately 41 kDa, with higher density in BA than MCA, indicating that protein levels of vasodilator prostanoid enzymes and their receptors are higher in BA than in MCA.

The protein levels of neuronal and endothelial NOS enzymes are similar between BA and MCA arteries

To evaluate if NO plays a role in the flow-induced dilation response of BA artery, we determined the protein levels of nNOS and eNOS isoenzymes (Figure 2C,D). We identified bands with similar density in the BA compared to the MCA for nNOS at approximately 166 kDa (Figure 2C) and endothelial eNOS at approximately 133 kDa (Figure 2D), respectively, suggesting that protein levels of these enzymes in BA and MCA are similar.

### 3.2. Gene Expression of Enzymes and Receptors of AA Cascade in Cerebral Arteries of PCC and ACC

We found that expressions of COX1 and COX2 genes are significantly lower in CA_PCC_ compared to CA_ACC_, suggesting that gene and protein expression patterns of COX1 and COX2 are correlated. In contrast, the expressions of Tbxas, Tbxa2r, Ptgis, Ptgir, Cy4a1, NOS1, and NOS3 genes were similar in CA_PCC_ compared to CA_ACC_. This suggests that gene expression and protein levels of these enzymes and receptors are not correlated.

#### 3.2.1. Expression of Multiple Genes Involved in Vascular Tone Regulation Is Different in Cerebral Arteries of PCC and ACC

To further evaluate if other mediators of vascular tone regulation are expressed differentially in cerebral arteries of PCC (such as BA) and ACC (such as MCA), we studied the gene expression at a transcriptome level by RNA-sequencing.

The DESeq analysis revealed that 959 genes had significantly different expression (adj *p* < 0.05) in CA_ACC_ relative to CA_PCC_. Ingenuity Pathway Analysis (IPA) was used to assess the functional enrichment analysis of genes, which included 636 genes (362 upregulated, 274 downregulated) that were differentially expressed (DEGs) in CA_ACC_ relative to CA_PCC_ (Figure 3).

Ingenuity Pathway Analysis revealed that 636 DEGs impacted 148 canonical pathways in a significant manner, from which 21 canonical pathways are involved in vascular tone regulation (Figure 4, Table 1). We assessed the predicted activity of these pathways, calculated as Z-score, based on the gene expression changes in CA_ACC_ relative to CA_PCC_ samples. Among the vasoconstriction-related pathways, six demonstrated a more activated pattern and two had a negative Z score in CA_ACC_ relative to CA_PCC_ samples (Figure 4A, Table 1). On the other hand, the activity for the majority of pathways involved in vasodilation was predicted to be suppressed, with only the Protein Kinase A signaling pathway demonstrating slight activation (Figure 4B, Table 1). These results suggest that vasoconstrictor mechanisms are more dominant in CA_ACC_ than in CA_PCC_, while vasodilator mechanisms are more dominant in CA_PCC_ than in CA_ACC_.

#### 3.2.2. Gene Ontology (GO) Analysis of DEGs in Cerebral Arteries of PCC and ACC

GO analysis identified nine DEGs that are involved in positive regulation of vasoconstriction: Adra1a (Adrenoceptor Alpha 1A), arginine vasopressin (Avp), epidermal growth factor receptor (Egfr), and histamine receptor H2 (Hrh2); vasodilation: Mas1 (Mas1 oncogene), Sod1 (superoxide dismutase 1), Npr3 (natriuretic peptide receptor 3), Kcnma1 (potassium calcium-activated channel subfamily M alpha 1), Abcc9 (ATP Binding Cassette Subfamily C Member 9), Vegfa (vascular endothelial growth factor A), and Itga1 (Integrin Subunit Alpha 1) (Table 2). Previous data indicated that these genes encode mediators of flow/wall shear stress-induced vasoconstriction or vasodilation (Table 2). These results suggest that, with the exception of P2rx1, Npr3 and Vegfa, the expression of vasodilator genes is more dominant in CA_PCC_ relative to CA_ACC_, while vasoconstrictors, with the exception of Egfr, are more dominant in CA_ACC_ relative to CA_PCC_.

## 4. Discussion

The salient findings of the present study are that, by and large:(1)The levels of enzymes and receptors involved in the production and action of arachidonic acid constrictor metabolites are lower in the basilar artery (BA) than in the intracranial middle cerebral artery (MCA);(2)The level of enzymes responsible for producing dilator mediators and receptors (such as PGIS and IP receptors) is greater in BA than in MCA;(3)The expression of nearly 1000 genes varies between cerebral arteries of the posterior (CA_PCC_) and anterior cerebral circulation (CA_ACC_);(4)The expression of 636 genes involved in the regulation of canonical pathways of flow-dependent vascular tone differs between the cerebral arteries of the PCC and ACC;(5)Vasodilation-related canonical pathways are more prominent in CA_PCC_ compared to CA_ACC_, while vasoconstriction-related canonical pathways are more prominent in CA_ACC_ compared to CA_PCC_;(6)The expression of nine genes involved in flow-dependent vasodilation and vasoconstriction differs between the CA_PCC_ and CA_ACC_. Specifically, vasodilator genes are predominantly upregulated in the CA_PCC_ relative to CA_ACC_.

Overall, these findings explain why a flow-induced dilation in the cerebral arteries of the posterior cerebral circulation (such as the BA) and flow-induced constriction in the cerebral arteries of the anterior cerebral circulation (such as the MCA) were primarily observed in previous studies, thereby providing a mechanistic basis for functional findings.

### 4.1. Physiological Implications of Our Findings: BA vs. MCA

To maintain optimal blood flow to the brain, supporting and preserving its function is of utmost importance. Coupling the vasomotor function of cerebral arteries of the posterior and anterior cerebral circulations serves this function. There have been several previous observations showing that cerebral arteries of the posterior and anterior cerebral circulations behave differently.

The large conduit BA directly supplies the brainstem and cerebellum, and through the circle of Willis, the posterior cerebral circulation. The research group of Faraci (1991) observed significant increases in the diameter of the rat BA during common carotid artery occlusions [7]. They interpreted this response as reactive hyperemia. Decreases in perfusion pressure in the circle of Willis (i.e., during carotid occlusion) result in decreases in cerebral vascular resistance (decrease in myogenic tone) in the MCA, which subsequently leads to increased diameter and blood flow in the BA, in part, by flow-induced dilation. Indeed, in the present study, we showed that potent vasodilator enzymes with higher protein levels are strongly present in the BA compared to MCA (Figure 2A,B), and thus, they may be responsible for producing dilator mediators and receptors (such as prostacyclin and its IP receptor).

One of the important functions of the cerebral arteries responsible for the anterior circulation (such as the MCA) is to contribute to the autoregulation of cerebral blood flow (CBF), i.e., to maintain CBF despite the changing perfusion pressure and flow. Also, the tight control of CBF is crucial not only to maintain the constant blood supply of the brain but also to maintain a relatively constant blood volume to comply with the limited space available in the skull [53]. Thus, cerebral arteries of the anterior circulation have a well-developed constrictor function. 

On the other hand, cerebral arteries of the posterior circulation (such as the BA) are not affected by these limitations, allowing them to dilate, which then increase the blood flow to the circle of Willis, where pressure and flow can equilibrate and provide an even supply of each artery branching off from it.

### 4.2. Autoregulation of CBF and Flow-Dependent Responses

Autoregulation of CBF is achieved, in addition to pressure-induced vasoconstriction [54], by the flow-sensitive function of the cerebral arteries of the anterior cerebral circulation, such as the MCA and penetrating arterioles, which constrict when intraluminal flow increases [4,10,12,55]. It is of note that, depending on the anatomical location, surrounding tissues and type of cerebral artery studied, dilation, constriction, or biphasic response were reported to increase in flow [7,10,11,55,56,57,58,59,60]. For example, previous studies have shown that increases in flow elicit dilations in the BA, which is located outside the cerebrum in the groove of the midbrain, allowing space and permitting dilation due to the fact that it is located before the circle of Willis [1,7,8].

The mediators involved in flow-induced vasomotor responses of cerebral vessels include primarily arachidonic acid (AA) metabolites, whereas nitric oxide seems to play a modulatory role. Thus, in the present study, we hypothesized that enzymes that are responsible for the production of vasomotor molecules involved in flow-dependent responses are expressed differentially in the BA and the MCA. To reveal the molecular background of the different vasomotor responses of cerebral arteries of the anterior and posterior cerebral circulations, in the present study, we measured the protein expression of key vasoactive AA enzymes and receptors, and NO synthases in the MCA and the BA.

### 4.3. Expressions of Enzymes Producing AA Metabolites and Their Receptors in the MCA and the BA Correspond to Their Vasomotor Responses to Flow

In the present study, we have found that (1) protein levels of enzymes in the AA pathway producing vasoconstrictor and vasodilator prostanoids are different in the MCA and the BA; (2) protein levels of COX1 and 2, which are involved in the production of vasoconstrictor and vasodilator prostanoids, and cytochrome P450 producing vasoconstrictor 20-HETE, are significantly higher in the MCA than in the BA; (3) the protein level of prostacyclin synthase (PGIS), which produces the vasodilator PGI2, is significantly higher in the BA than in the MCA; (4) the protein level of thromboxane synthase (TXAS) producing vasoconstrictor TXA2 is similar between the MCA and the BA; (5) the protein level of thromboxane A2 receptor is significantly higher in the MCA than in the BA and protein level of prostacyclin receptor is significantly higher in the BA than in the MCA; and (6) the protein levels of neuronal and endothelial NOS (nNOS and eNOS) enzymes are similar between the MCA and the BA.

Sensing flow is an important feature of vessels, which was investigated in great depth by several investigators [8,9,12,55,60,61,62,63,64,65]. These studies revealed that the signaling of flow-induced vasomotor responses of cerebral vessels includes glycocalyx, ion channels, integrins, extracellular matrix molecules, cytoskeleton, and second messengers [4,8,9,37,61,64,65,66,67,68]. Among second messengers, vasoactive metabolites of the AA pathway have been shown to mediate mechano-signaling in cerebral vessels [4,12]. AA is a major component of the cell membrane and precursor of several vasoactive prostanoid molecules with important roles in the regulation of vascular tone [13]. It is envisioned that increases in flow activate phospholipase A2 (PLA2), eliciting the release of AA from the cell membrane [66].

AA is metabolized via three enzymatic pathways: the cyclooxygenase (COX), lipoxygenase, and cytochrome P450 (CYP450) pathways [15,69]. COX enzymes, such as COX1 and COX2, metabolize AA into the endoperoxide prostaglandin H2 (PGH2), which is a precursor for the enzymes thromboxane synthase (TXAS) and prostaglandin synthase (PGIS). PGH2 can be converted to thromboxane A2 (TXA2) by TXAS or to prostacyclin (PGI2) by PGIS [66]. TXA2 binds to contractile G protein (Gq)-coupled thromboxane-prostanoid receptor (TP receptor, TBXA2R) that mediates Ca^2+^ mobilization in the VSMC and induces VSMC contraction and vasoconstriction. PGI2 binds to the Gs-coupled prostacyclin receptor (IP), inducing VSMC relaxation through an increase of cyclic adenosine monophosphate (cAMP) [70]. In vivo and in vitro studies showed that increasing flow elicited constriction in the isolated rat MCA, whereas dilation occurred in the BA [4,7,8,55]. It has been shown that flow-induced vasoconstriction responses of the isolated rat MCA were attenuated by COX inhibitor indomethacin and TP receptor inhibitor SQ 29,548, suggesting a role of COX metabolites [4].

In the present study, we found that levels of COX1, COX2, and TP receptor proteins were significantly higher in the MCA compared to BA (Figure 1A,B,D). It should be noted that Figure 1A,B show COX2 protein levels twice as high as COX1, suggesting a greater role for COX2, as shown by others and thus perhaps explaining the unexpected negative effects of some of the specific COX2 inhibitors on the cerebral circulation [71,72].

Furthermore, Toth et al. (2011), using ozagrel to block TXA2 production, found that the flow-induced vasoconstrictor responses of the isolated rat MCA were not affected [4]. In the present study, we found no difference in the TXAS levels between the MCA and the BA arteries (Figure 1C), which is in line with the findings of Toth et al., 2011 [4]. However, we also found that the protein level of the TP receptors was significantly greater in the MCA than in the BA arteries (Figure 1D), which suggests that TXA2 is likely responsible for the opposite vasomotor responses of the MCA and BA to flow.

CYP450 enzymes catalyze epoxidation reactions (producing epoxyeicosatrienoic acids, EETs) and omega-hydroxylation reactions (producing hydroxy-eicosatetraenoic acids, HETEs) [70]. CYP4A subfamily hydroxylase enzymes are responsible for 20-HETE production in the cerebral circulation [13]. In rats, the CYP4A3 isoform is predominantly found in the cerebral vasculature; however, other isoforms are also present, including CYP4A1, CYP4A2, CYP4A8, and CYP4F [19,69]. 20-HETE exerts its vasoconstriction action through several mechanisms, including acting on the newly discovered Gq-coupled receptor G-protein-coupled receptor 75 (GPR75) [69]. In the present study, we found that the protein level of the CYP4A enzyme is significantly greater in the MCA compared to BA, suggesting that the 20-HETE is likely responsible for the flow-induced vasoconstriction responses of the MCA (Figure 1E).

#### PGI2 and NO: Mediators and/or Modulators of Flow-Induced Responses in Cerebral Arteries

Flow-induced responses of cerebral vessels are likely mediated and/or modulated by NO and vasodilator prostaglandins [8,11,37,73]. Nevertheless, Fuji et al. (1991) found in vivo that the topical administration of N(G)-monomethyl-L-arginine (L-NMMA, a known L-Arg analog that competitively inhibits NOS) did not reduce the flow-induced dilation of the rat BA, which was initiated by occlusion of the common carotid artery [7]. In contrast, Paravicini et al. (2006), using the same approach, observed a significant reduction in flow-dilation of the BA by topical administration of another NOS inhibitor, N(G)-nitro-L-arginine methyl ester (L-NAME) [8]. Likewise, Lieberman et al. (1996) showed that the in vivo intra-arterial administration of L-NMMA inhibited the flow- and acetylcholine-induced dilations, respectively, in the human brachial artery [74]. In the intracranial anterior and posterior cerebral arteries of mice, Drouin and Thorin (2009) demonstrated that L-NAME inhibited the flow-induced dilation [62]. It was also shown that bradykinin, known to release NO and PGI2, elicited dilation of the isolated rat MCA and BA, respectively [73].

In the present study, we found that the levels of PGIS enzyme (producing PGI2) and PTGIR receptor were significantly higher in the BA compared to MCA (Figure 2A,B); however, we did not find difference in the levels of nNOS and eNOS in the MCA and BA (Figure 2C,D).

These findings suggest that NO, together with other vasomotor mediators, contributes to the vasomotor responses of cerebral arteries to hemodynamic forces. In the BA, it is a primary mediator [8,75,76], whereas in the MCA, it is a modulator of the response [11,57,60,73].

This idea is supported by the findings of Roman’s laboratory, which showed that while the dilator responses of the rat BA are guanylate cyclase/cyclic guanosine monophosphate (GC/cGMP) dependent, in the MCA, GC accounts for only ~50–25% of the dilator responses to NO donors and the rest is mediated by other mechanisms, involving NO-induced inhibition of 20-HETE and activation of calcium-activated K^+^-channels [77]. Our study’s results showed that the level of the CYP4A enzyme is higher in the MCA than in the BA (Figure 1E). However, the levels of nNOS and eNOS proteins are similar in the MCA and the BA (Figure 2C,D), which supports the previous functional findings that 20-HETE contributes to the constriction responses in the MCA, while NO contributes to the dilation responses in the BA to increasing intraluminal flow. Overall, the cellular protein levels of the enzymes producing vasoactive AA metabolites, such as COX1/2, CYP4A, PGIS, and NOS, correspond to the opposite flow-induced vasomotor responses in the MCA and the BA.

### 4.4. Genes Regulating Enzyme Expression, Producing Mediators of Vascular Tone Show Differential Expression Patterns Between Cerebral Arteries of the PCC and the ACC

#### 4.4.1. Expression of Genes of Key Enzymes and Receptors of the AA Pathway and NO Synthases

In the present study, we found that, similar to protein levels, the expression of COX1 and COX2 enzyme genes was higher in the CA_ACC_ relative to CA_PCC_, and expression of the TXAS enzyme gene was similar in the CA_ACC_ and the CA_PCC_. However, contrary to protein levels, the expressions of the PGIS and Cyp4A genes, as well as the PTGIR and TP genes, were similar in the CA_ACC_ and the CA_PCC_. This result indicates that protein expression of certain AA pathway molecules, such as the Cyp450 enzymes and TP receptors, is not proportional to their gene expression. However, one study found that while some cytochrome P450 enzymes showed similar patterns of gene and protein expression, others did not [78]. For example, Cyp2E showed high gene expression in the intestine with no corresponding protein expression, indicating post-translational interference. Another study found that while the IP receptor mRNA was highly expressed during certain phases of the menstrual cycle within the human endometrium, the immunoreactive IP receptor protein showed different expression patterns [79]. These findings suggest that with the exception of COX1 and COX2, where protein expression matches gene expression, the protein expression of AA-related enzymes and receptors is not directly proportional to gene expression.

One can propose that hemodynamic forces (such as shear stress) can act as epigenetic factors and, in effect, act as a post-translational mechanism. Indeed, a recent article by Aradhyula et al. published in 2024 discussed the potential role of hemodynamic forces as epigenetic factors modulating AA genes translation, including Cyp450 [80]. In addition, a study by Jiang et al. in 2015 provided a comprehensive overview of how flow-mediated epigenetic responses can regulate endothelial function, showing that local hemodynamic forces influence endothelial phenotype through epigenetic mechanisms, including post-translational modifications [81]. Finally, another study by Russel-Pureli, demonstrated that fluid shear stress activates COX2 and PGI2 release in endothelial cells [82].

In the present study, we also found that expressions of NOS1 and NOS3 genes encoding neural and endothelial NOS enzymes were similar between CA_PCC_ and CA_ACC_. Our finding indicates that mRNA and protein expression levels of neural and endothelial NOS are similar in both CA_PCC_ and CA_ACC_, suggesting that the regulatory mechanism governing their expression is consistent across these cerebral vascular regions, despite the differences in hemodynamic forces and other local factors. A review article by Costa et al. in 2016 highlighted that both nNOS and eNOS are constitutively expressed in various vascular regions, including cerebral arteries [83], which correspond to vasomotor function as ACh elicits dilations in both vasculatures [8,11,57,60,73,75,76].

#### 4.4.2. Expression of Multiple Genes Involved in the Canonical Pathways of Vascular Tone Regulation Differs in Cerebral Arteries of the PCC and ACC

In the present study, we found that nearly 1000 genes were expressed differently in cerebral arteries of the PCC and ACC. Among these genes, 636 genes were found to be involved in the regulation of 148 canonical pathways, from which 21 canonical pathways are involved in the flow/WSS-dependent vascular tone regulation (Figure 4, Table 1). Canonical pathways of vasoconstriction G alpha (q) signaling events; Gαi Signaling; G alpha (12/13) signaling events; Endothelin-1 Signaling; and Signaling by Rho Family GTPases have been shown to be regulated by WSS (for reference, see Table 1). We found that genes regulating G alpha (q) signaling events and G alpha (12/13) signaling events are mostly upregulated in the CA_ACC_ relative to the CA_PCC_; however, genes regulating other vasoconstrictor pathways are mostly downregulated in the CA_ACC_ relative to the CA_PCC_ (Table 1).

This result suggests that the dominance of G alpha (q) signaling events and G alpha (12/13) signaling events in the CA_ACC_ may be contributing to the flow-induced vasoconstriction in the MCA. Canonical pathways of vasodilation, including Estrogen Receptor Signaling, Gαs Signaling, Acetylcholine Receptor Signaling Pathway, Gαβ Signaling, Potassium Channels, Protein Kinase A Signaling, Nitric Oxide Signaling in the Cardiovascular System, and cAMP-mediated signaling, have been shown to be regulated by WSS (for references see Table 1). We found that, except for Gαs Signaling, the genes of vasodilator canonical pathways are downregulated in CA_ACC_ relative to the CA_PCC_. This finding may explain why increases in blood flow induce vasodilation in the basilar artery.

#### 4.4.3. Multiple Genes Regulating Flow-Dependent Vascular Tone Show Differential Expression Patterns Between Cerebral Arteries of the Posterior and Anterior Cerebral Circulation

The genes Egfr, P2rx1, Npr3, Kcnma1, Mmp28, Abcc9, Vegfa, Itga1, and Itga9 have been shown to be involved in the flow-mediated vascular tone regulation (for references, see Table 2). In the present study, we found that the vasodilator genes, such as Kcnma1, Mmp28, Abcc9, Itga1, and Itga9, are downregulated in the CA_ACC_ relative to the CA_PCC_, which may be responsible for the flow-dependent vasodilation in the basilar artery. Interestingly, we found that the vasoconstrictor gene Egfr is downregulated, and the vasodilator genes P2rx1, Npr3, and Vegfa are upregulated in the CA_ACC_ relative to the CA_PCC_. This finding suggests that although the expression of vasodilator genes is more prominent in the CA_PCC_ and the vasoconstrictor genes are more prominent in the CA_ACC_, these differences may underlie the flow-induced vasodilation observed in the CA_PCC_, such as the basilar artery, and the flow-induced vasoconstriction observed in the CA_ACC_, such as the middle cerebral artery.

Overall, multiple genes exhibit differential expression patterns in the cerebral arteries of the PCC and the ACC (see Table 1 and Table 2). Data from previous studies indicate that these genes play a role in the pressure and flow-induced vascular tone regulation.

By integrating previous data into our findings, we enhanced the strength of our conclusions by enabling direct comparison in the ‘Results’ section, thereby increasing the relevance and reliability of our findings.

### 4.5. Physiological Importance and Possible Clinical Applications of Findings

Collectively, the findings of the present study suggest that the differential expression of vasoactive enzymes and receptors of the arachidonic acid (AA) pathway corresponds to the opposite vasomotor functions of cerebral arteries of the posterior and the anterior cerebral circulations. We propose that the greater expressions of vasodilator enzymes and receptors in the BA lead to dilation, whereas greater expressions of vasoconstrictor enzymes and receptors in the MCA lead to constriction in response to flow.

The opposite vasomotor responses of the BA and MCA are crucial for an equilibrated blood supply of the brain, but at the same time, they must regulate cerebral blood volume. The impairment of these mechanisms can be particularly important in cerebrovascular diseases, such as cerebral ischemia, traumatic brain injury and hypertension, and preeclampsia, when loss of intracranial arterial constriction may lead to increased perfusion pressure in the distal segment of the microcirculation resulting in increased blood–brain barrier permeability, edema, and hemorrhagic stroke [84,85,86]. On the other hand, loss of dilation in the arteries of the posterior cerebral circulation, such as the BA, may lead to decreased blood supply to the brain, resulting in ischemic stroke.

In the present study, we utilized the basilar artery of the vertebrobasilar system, which is not affected by surrounding space limitation, and the middle cerebral artery (MCA) of the internal carotid artery system, which is deeply embedded in the brain, where the surrounding space is limited. These special localizations likely determine their vasomotor responses to various biological mediators and hemodynamic forces, and consequently gene and protein expressions.

The arteries of the anterior cerebral circulation are susceptible to atherosclerosis, which can cause flow-limiting stenosis and embolization of plaque distally towards the brain [87]. Embolization of atherosclerotic plaque within the internal carotid artery predominantly obstructs the middle cerebral artery, leading to ischemic stroke. In such cases, compensatory basilar artery dilation could facilitate increased blood flow towards the middle cerebral artery via collateral pathways in the circle of Willis, thereby mitigating the effects of arterial blockage [88]. Redistributing collateral blood flow via vasodilation in the BA (to the posterior cerebral circulation) to enhance collateral circulation through the circle of Willis and reducing vasoconstriction in the MCA could compensate for the reduction of blood flow in the MCA during acute ischemic events (Figure 5).

Our findings include several therapeutic, diagnostic, and medication possibilities. Administration of pharmacological vasodilators based on prostaglandins (e.g., prostaglandin analogs, such as epoprostenol) could enhance the vasodilator capacity of the BA. The inhibition of vasoconstrictor eicosanoids, such as TXA2, by administration of thromboxane receptor antagonists could attenuate vasoconstriction during ischemic periods. The determination of AA-derived metabolites in the cerebrospinal fluid or blood plasma as biomarkers for imbalanced vasoconstriction and vasodilation could be a diagnostic tool. AA-derived drugs, such as prostacyclin analogs (e.g., iloprost and treprostinil) to enhance vasodilation in the BA, or TXA2 receptor antagonists (e.g., seratrodast) to attenuate vasoconstriction in the MCA, or using selective modulators of prostaglandin receptors (e.g., EP4 agonists) for vascular relaxation, and COX inhibitors to reduce vasoconstrictor metabolites while preserving vasodilator prostaglandins can be used in the medication of stroke patients.

Another possibility is endarterectomy of stenosis in the carotid artery, which likely involves the capacity of the basilar artery to dilate in response to increasing flow, enhancing the inter-hemispheric perfusion through the circle of Willis, thereby improving the cerebral circulation [89,90].

In conclusion, the novel findings of the present study are that (1) The expressions of enzymes and receptors involved in the production and action of arachidonic acid constrictor metabolites are greater in the middle cerebral artery than in the basilar artery, (2) The expression of enzymes responsible to produce dilator mediators (such as NO) are similar in the MCA and BA, and (3) The expression of multiple genes involved in vasodilator mechanisms is more prominent in the cerebral arteries of the posterior, while vasoconstrictor mechanisms are more prominent in the cerebral arteries of the anterior cerebral circulation, with several of these genes governing flow-dependent mechanisms.

As depicted in Figure 6, these findings may explain the different flow-dependent vasomotor responses of the MCA and BA observed in previous studies, i.e., constriction in the MCA and dilation in the BA, whereas the similar expression of enzymes producing dilator molecules supports their important modulatory roles. Nevertheless, additional signaling mechanisms are likely contributing to the flow-dependent responses of cerebral vessels, which need to be revealed by future studies.

## Figures and Tables

**Figure 1 life-15-00856-f001:**
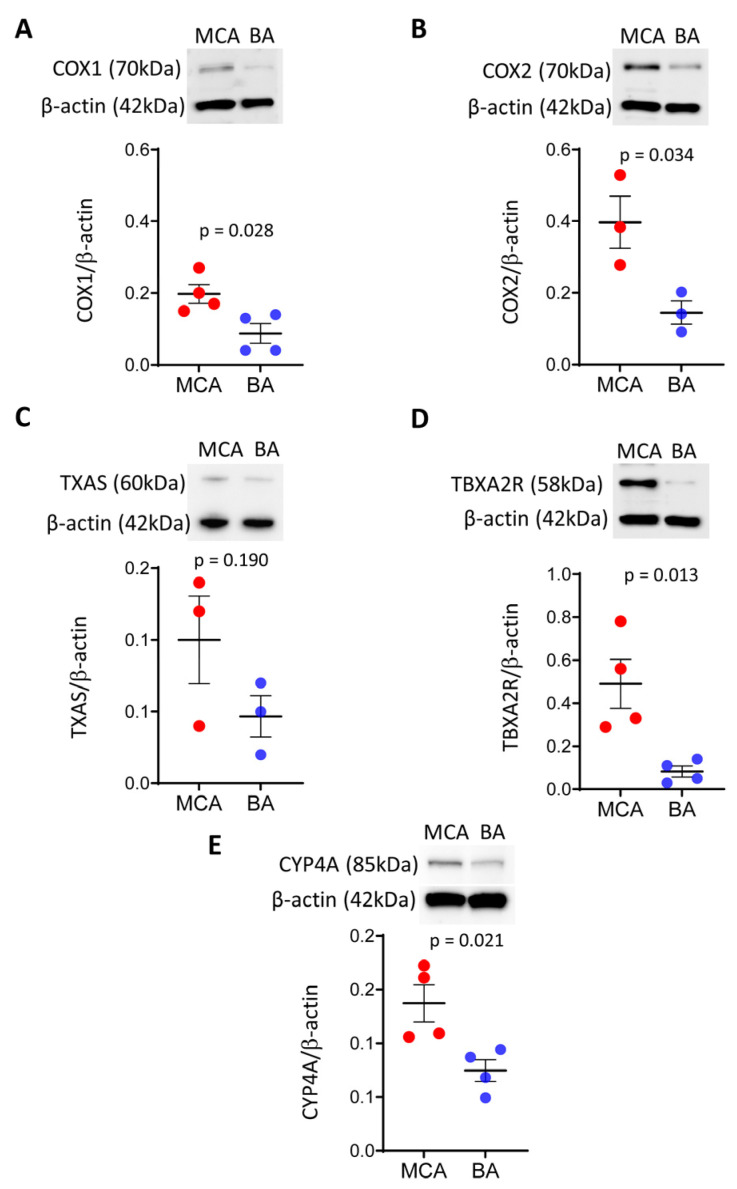
Protein levels of vasoconstrictor enzymes COX1/2, CYP4A and TP receptors are significantly lower in BA, while the protein level of TXAS is similar between BA and MCA arteries. Representative Western blots and group data of densitometric analysis of COX1 (**A**) and COX2 (**B**), TXAS (**C**), TP receptor (**D**), and CYP4A (**E**) levels are shown. Results were normalized to β-actin levels. Comparisons were made by parametric two-tailed independent *t*-tests. Data are expressed as mean ± SEM of three individual samples (six animals/sample) from 1–2 separate trials. *p* < 0.05 was considered statistically significant. BA—basilar artery; MCA—middle cerebral artery; COX1/2—cyclooxygenase 1 and 2; TP receptor/TBXA2R—thromboxane A2 receptor; TXAS—thromboxane synthase; CYP4A—Cytochrome P450 4A.

**Figure 2 life-15-00856-f002:**
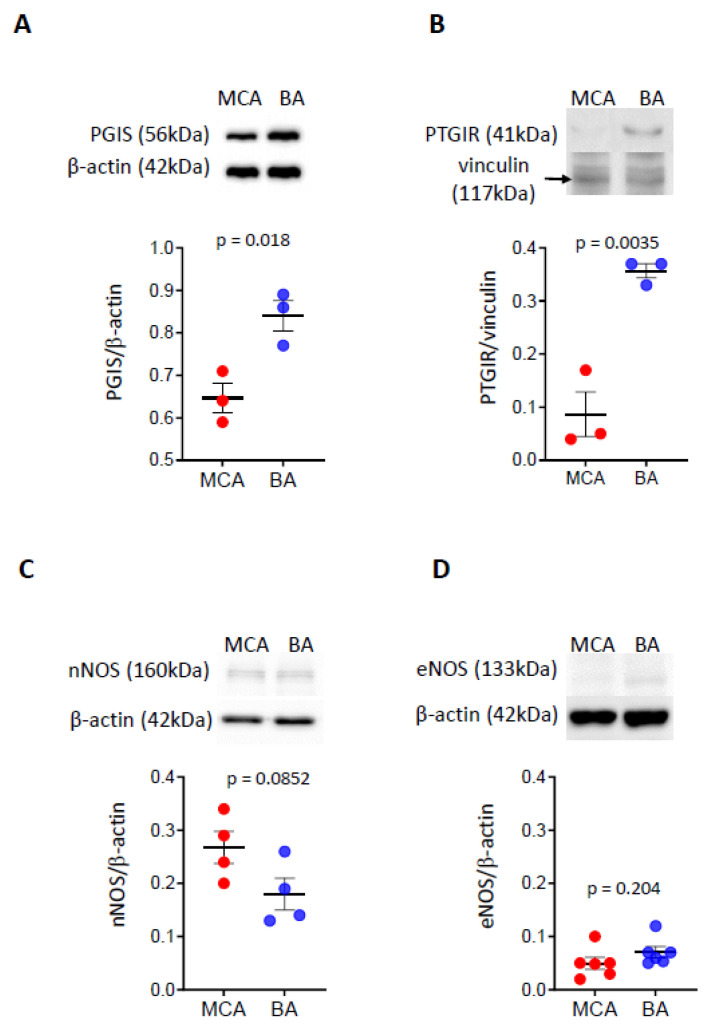
Protein levels of prostacyclin synthase and prostacyclin receptor, as well as neural and endothelial NOS enzymes in BA and MCA. The levels of PGIS and PTGIR proteins are significantly higher in BA than MCA (**A**,**B**). The levels of nNOS and eNOS proteins are similar in BA and MCA (**C**,**D**). Representative Western blots and group data of densitometric analysis of PGIS (**A**), PTGIR (**B**), neural NOS (**C**), and endothelial NOS (**D**) levels are shown. The results were normalized to β-actin levels. Comparisons were made by parametric two-tailed independent *t*-tests. Data are expressed as mean ± SEM of three individual samples (six animals/sample) from 1–2 separate trials. *p* < 0.05 was considered statistically significant. BA—basilar artery; MCA—middle cerebral artery; PGIS—prostacyclin synthase; PTGIR—prostacyclin receptor; nNOS—neural nitric oxide synthase; eNOS—endothelial nitric oxide synthase.

**Figure 3 life-15-00856-f003:**
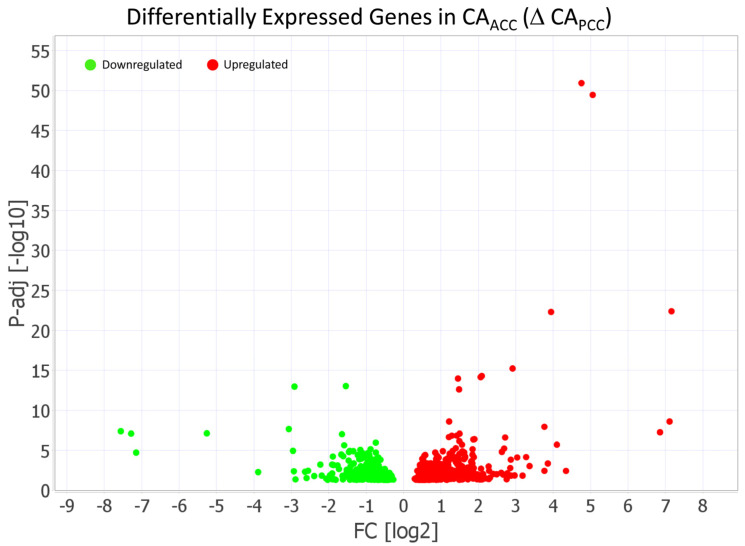
Volcano plot demonstrating 636 differentially expressed genes (DEGs) in cerebral arteries of anterior (CA_ACC_) relative to posterior (CA_PCC_). Ingenuity Pathway Analysis (IPA) identified 362 differentially expressed genes (DEGs) that were upregulated and 274 DEGs that were downregulated in CA_ACC_ relative to CA_PCC_. The light green dots represent downregulated genes (FC < −1.5), while the red dots represent upregulated genes (FC > 1.5) in CA_ACC_ relative to CA_PCC_. The threshold of the FDR-adjusted *p*-value was set <0.05.

**Figure 4 life-15-00856-f004:**
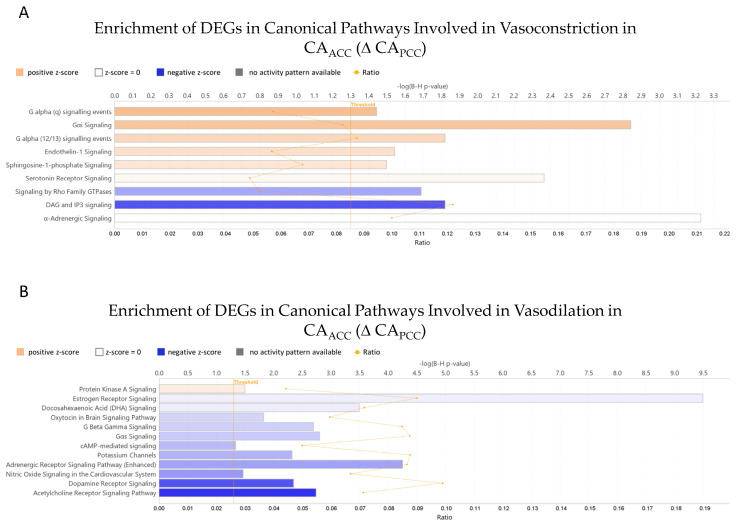
Ingenuity pathway analysis (IPA) of differentially expressed genes of canonical pathways involved in vasoconstriction (**A**) and vasodilation (**B**) in CA_ACC_ relative to CA_PCC_. Selected canonical pathways are grouped by their proposed effect on vascular tone regulation and presented accordingly. Adjusted *p*-values of pathway enrichment are visualized on a −(lg) scale, where the values above 1.3 equal the threshold of significance below 0.05. The ratio value is calculated by dividing the number of DEGs mapped to the pathway by the number of all the genes present on the given canonical pathway. The coloring of the bars represents the predicted activity of the canonical pathways calculated as Z-scores. DEGs—differentially expressed genes; CA_ACC_—cerebral arteries of the anterior cerebral circulation; CA_PCC_—cerebral arteries of the posterior cerebral circulation.

**Figure 5 life-15-00856-f005:**
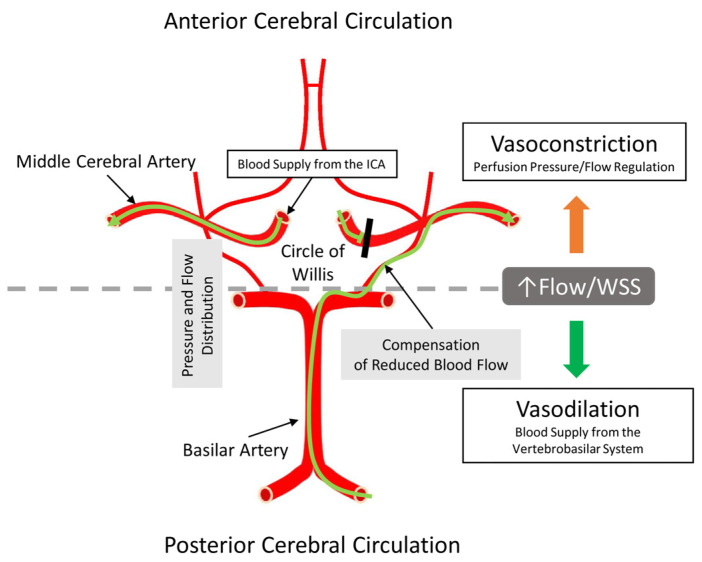
Illustration of the anatomical origin of arteries investigated in the present study. The important point is that on the base of the brain, the circle of Willis is located, is supplied by several arteries, and serves to equalize the pressure and flow in the circle of Willis before further daughter arterial branches reach the intracerebral spaces. Because of this special arrangement, the circle of Willis can provide compensation and maintenance of blood supply to the anterior brain regions if one of the supplying arteries is stenotic or blocked. ICA—internal carotid artery; WSS—wall shear stress. The green line with an arrowhead indicates the direction of blood flow. The black line indicates blocked blood flow through the middle cerebral artery.

**Figure 6 life-15-00856-f006:**
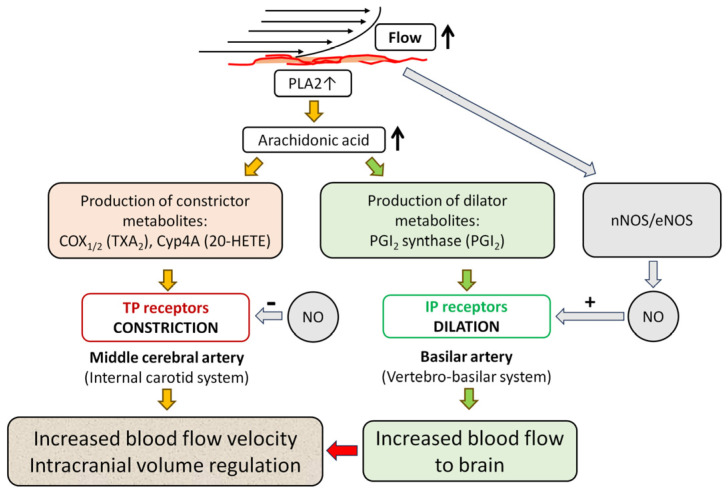
Proposed molecular mechanisms of the opposite vasomotor responses of the middle cerebral and basilar arteries to flow. An increase in intraluminal flow elicits the release of arachidonic acid (AA) from the cell membrane via activation of phospholipase A2 (PLA2). This flow elicits constriction in the internal carotid system, such as the middle cerebral artery (MCA), while in the vertebro-basilar system, such as the basilar artery (BA), the flow elicits dilation. We propose that different expressions of enzymes producing AA metabolites may be responsible for the opposite vasomotor responses. Expression of enzymes, thus the production of constrictor AA metabolites, is greater in the MCA than in BA, whereas expression of enzymes producing dilator AA metabolites is greater in BA than in MCA. Flow-induced constriction is mediated primarily by 20-HETE produced by cytochrome P450 4A (Cyp4A) enzymes and requires COX1/2 activity and thromboxane (TP) receptor, whereas the flow-induced dilation is mediated by dilator AA metabolites, such as prostacyclin (PGI2) produced by PGI2 synthase (PGIS) and the PGI2 receptor (IP). Nitric oxide (NO) produced by NO synthases (nNOS/eNOS) in response to increases in flow acts as a modulator through attenuating constriction in MCA, while mediating dilation in BA. Flow-induced dilation in the BA increases blood flow to the brain, which leads to constriction in the MCA. Flow-induced constriction contributes to the autoregulation of cerebral blood flow (CBF) by intracranial volume regulation to maintain a constant CBF in the face of increased pressure and blood volume [4,12].

**Table 1 life-15-00856-t001:** Significantly Enriched Canonical Pathways Involved in Wall Shear Stress-Induced Vascular Tone Regulation in CA_ACC_ Relative to CA_PCC_.

Canonical Pathways Involved in WSS-Induced Vasoconstriction	No. of Up/Downregulated Genes in CA_ACC_ (D CA_PCC_)	References
G alpha (q) signaling events	8/2	[24,25]
Gαi Signaling	5/7	[25,26]
G alpha (12/13) signaling events	5/2	[24,25]
Endothelin-1 Signaling	3/8	[27,28]
Signaling by Rho Family GTPases	5/9	[29,30,31]
**Canonical Pathways Involved in Vasodilation**		
Estrogen Receptor Signaling	7/13	[32]
Gαs Signaling	7/6	[25,26]
Acetylcholine Receptor Signaling Pathway	2/12	[33]
Gαβ Signaling	4/7	[25,34]
Potassium Channels	3/6	[35]
Protein Kinase A Signaling	7/11	[36]
Nitric Oxide Signaling in the Cardiovascular System	3/5	[37]
cAMP-mediated signaling	5/7	[38,39]
**Canonical Pathways Involved in Vasoconstriction and Vasodilation**		
Eicosanoid Signaling	7/10	[4]
Dopamine Receptor Signaling	3/5	[40,41]
Sphingosine-1-phosphate Signaling	1/2	[25]

WSS—wall shear stress; CA_AAC_—cerebral arteries of the anterior cerebral circulation; CA_PCC_—cerebral arteries of the posterior cerebral circulation.

**Table 2 life-15-00856-t002:** Gene ontology enrichment of differentially expressed genes related to wall shear induced vasomotor responses in CA_ACC_ Relative to CA_PCC_.

Gene	Up (+) or Downregulated (−) in CA_ACC_ (D CA_PCC_)	Flow/WSS-Induced Vasomotor Response	Reference
Egfr	−	Vasoconstriction	[42]
P2rx1	+	Vasodilation	[43,44]
Npr3	+	Vasodilation	[45,46]
Kcnma1	−	Vasodilation	[47]
Mmp28	−	Vasodilation	[48]
Abcc9	−	Vasodilation	[49]
Vegfa	+	Vasodilation	[49,50]
Itga1	−	Vasodilation	[51,52]
Itga9	−	Vasodilation	[51,52]

WSS—wall shear stress; CA_ACC_—cerebral arteries of the anterior cerebral circulation; CA_PCC_—cerebral arteries of the posterior cerebral circulation.

## Data Availability

Data are available within the article and Appendix A.

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
