# Peer review of "Differential Gene and Protein Expressions Responsible for Vasomotor Signaling Provide Mechanistic Bases for the Opposite Flow-Induced Responses of Pre- and Post-Circle of Willis Arteries"

_life, 2025, doi:10.3390/life15060856_

Round 1
Reviewer 1 Report
Comments and Suggestions for Authors
This manuscript investigates the differential expression of mediators involved in flow-dependent responses in extracranial (ExCA) and intracranial arteries (InCA). Key enzymes in the arachidonic acid (AA) pathway, including neuronal nitric oxide (NO). which serve as mediators of flow-dependent vasomotor signaling, exhibit distinct expression patterns in extracranial and intracranial arteries. But I still have the following problems to solve.
- What was the rearing environment for the mice, and what were the baseline conditions (e.g., body weight) at the time of sacrificed?
- Operating Environment: What were the conditions of the operating environment during sacrificed, and in what setting were the brain tissues and arteries of the rats dissected?
- Were the rats designated for Western blot analysis and RNA-seq analysis sacrificed simultaneously?
- Does the anti-CYP4A11 used adequately reflect Cyp450 expression levels?
Author Response
Dear Prof. Dr. Smania,
Thank you for serving as the editor on our manuscript (ID: life-3571395) entitled "Differential gene and protein expressions responsible for vasomotor signaling provide mechanistic bases for the opposite flow-induced responses of extra - and intracranial arteries". We are grateful for the helpful comments and suggestions of the Reviewers. Each comment and our response are provided in the following section. All modifications are indicated in red text in the revised manuscript.
Authors Introduction to Reviewers Comments:
The authors would like to thank the Reviewers for their thoughtful and insightful comments. We addressed each comment point by point in the following section. In response to these comments, we have amended our revised manuscript. We feel that these modifications have substantially improved the revised manuscript.
Responses to Reviewer 1 comments:
Comments 1: What was the rearing environment for the mice, and what were the baseline conditions (e.g., body weight) at the time of sacrificed?
Response 1: We agree with this comment. The rats were housed in groups with three rats/cage under controlled temperature (25±2 °C) and constant light cycle (12 hr light/dark) and allowed free access to a standard rat chow diet and water. Rats were purchased at 3 months of age. The rats underwent an acclimation period of 3-7 days before being used in experiments. At the time of sacrifice, their body weight ranged from 250-350 grams, consistent with the previous study by Toth et al. 2011 (PMID: 21610722) cited in the original manuscript. Decapitation was performed in an isolated area to prevent the rats from witnessing each other's decapitation, ensuring ethical and physiological/behavioral considerations.
This information - in brief - is now included in the revised MS in red fonts.
Comments 2: Operating Environment: What were the conditions of the operating environment during sacrificed, and in what setting were the brain tissues and arteries of the rats dissected?
Response 2: Thank you for the questions. Rats were sacrificed by decapitation using a rat guillotine following isoflurane anesthesia, ensuring deep anesthesia (e.g., slowed breathing). The operating room was isolated from other laboratory areas. Sacrifices were carried out individually, so rats did not witness each other's decapitation. Fur was removed from the head, and brains were extracted using bone cutter scissors and forceps. Brains were transferred into cold PBS in a petri dish. Surface cerebral arteries of the anterior cerebral circulation (circle of Willis and its branches and middle cerebral arteries), and basilar arteries were dissected with sterilized microsurgical tools (tweezers, scissors), removing all connective tissues (e.g., meninges) and brain tissues. Arteries were washed off from blood in cold PBS, and transferred into sterile Eppendorf tubes (arteries of the anterior cerebral circulation separated from basilar arteries), and transferred in liquid nitrogen, and then stored at -80°C until use. One common protocol was used for all tissue harvest procedures.
This information - in brief - is now included in the revised MS in red fonts.
Comments 3: Were the rats designated for Western blot analysis and RNA-seq analysis sacrificed simultaneously?
Response 3: Thank you for the question. No, they were not. The rats designated for Western blot and RNA-sequencing analyses were sacrificed at different time points, while maintaining consistent anesthetic protocols and vessel harvesting conditions.
This information is now included in the revised MS in red fonts (Page 3, lines 103-106).
Comments 4: Does the anti-CYP4A11 used adequately reflect Cyp450 expression levels?
Response 4: Thank you for the excellent question. No, it does not. The antibody used in our Western blot experiments, anti-CYP4A11 (Abcam ab3573, Anti-Cytochrome P450 4A/CYP4A11 antibody (ab3573) | Abcam), recognizes P450 enzymes within the 4A subfamily. However, the datasheet does not specify whether it specifically recognizes specific CYP4A isoforms. Cytochrome P4504A isoforms, such as CYP4A1, CYP4A3, and CYP4A8 contribute rat cerebral vascular 20-HETE production, thereby cerebral blood flow autoregulation (PMID: 21356152). We have inserted in red font the following “P450 4A/CYP4A11” in the revised manuscript (Page 3, lines 134-135): “P450 4A/CYP4A11”.
Reviewer 2 Report
Comments and Suggestions for Authors
The manuscript titled: “Differential gene and protein expressions responsible for vasomotor signaling provide mechanistic bases for the opposite flow-induced responses of extra - and intracranial arteries” is very interesting for readers. This paper fills the knowledge gap regarding molecular mechanism of the opposite vasomotor responses of the middle cerebral and basilar arteries to flow. The Authors used a different technique, e.g. Western blot, RNA sequencing analysis and bioinformatic analysis. The conclusions are correct.
As a Reviewer, I would like to make the followed comments:
- In paragraph: “Statistical analysis and calculations” is not mentioned the statistical test used to analysis (parametric or non-parametric tests)
- Results presentation is not well prepared and it need to be rewritten, e.g.:
- the numbering of figures is incorrect (on pages 5-7 the authors presented Figures 1-3, but on page 8 - Figure 1, what is contradictory with reference in the text (in line 289, page 7 is mentioned Figure 4B….???)
- The titles of Figures like as the descriptions of results, e.g. page 5, lines: 202-203 the Authors have written: “Expressions of vasoconstrictor enzymes COX1/2, CYP4A and TP receptor are significantly lower in BA, while expression of TXAS is similar BA and MCA arteries”, but in lines 198-200 was mentioned: We have found a significantly lower expression of COX1 and COX2 in BA compared to MCA arteries (Fig1A and B)”. Similarly, in lines 240-241 are mentioned the same information as in lines 233-235. Similar comments refer to Figure 3 on page 7.
- The numbering of tables (page 8-9) is incorrect. Additionally, in these tables, the Authors compared their results with literature data, which should not have happened in the Results section, but in Discussion. The comments dealing with the literature data (lines 316-317: Literature data indicated that these 9 genes encode mediators of WSS-induced vasoconstriction or vasodilation (Table 2)) should be moved to Discussion section.
Due to the above, the presentation of results is difficult to follow for it.
Author Response
Dear Prof. Dr. Smania,
Thank you for serving as the editor on our manuscript (ID: life-3571395) entitled "Differential gene and protein expressions responsible for vasomotor signaling provide mechanistic bases for the opposite flow-induced responses of extra - and intracranial arteries". We are grateful for the helpful comments and suggestions of the Reviewers. Each comment and our response are provided in the following section. All modifications are indicated in red text in the revised manuscript.
Authors Introduction to Reviewers Comments:
The authors would like to thank the Reviewers for their thoughtful and insightful comments. We addressed each comment point by point in the following section. In response to these comments, we have amended our revised manuscript. We feel that these modifications have substantially improved the revised manuscript.
Responses to Reviewer 2 comments:
“The manuscript titled: “Differential gene and protein expressions responsible for vasomotor signaling provide mechanistic bases for the opposite flow-induced responses of extra - and intracranial arteries” is very interesting for readers. This paper fills the knowledge gap regarding molecular mechanism of the opposite vasomotor responses of the middle cerebral and basilar arteries to flow. The Authors used a different technique, e.g. Western blot, RNA sequencing analysis and bioinformatic analysis. The conclusions are correct. As a Reviewer, I would like to make the followed comments:”
Comments 1: In paragraph: “Statistical analysis and calculations” is not mentioned the statistical test used to analysis (parametric or non-parametric tests)
Response 1: Thank you for pointing this out. We think explicitly mentioning “parametric test” is redundant, because “independent t-test” what we used for analyzing our protein expression data, and “Wald test” used in DESeq2 are mentioned in the “Statistical analysis and calculations”. The independent t-test is inherently a parametric test, which implies the assumptions of normality and equal variances. The Wald test, used in DESeq2, is also a parametric test assuming a specific distribution model for data. Nevertheless, we inserted the word “parametric test” in the description of statistical test in red font (Page 5, line 206).
.
Comments 2: “Results presentation is not well prepared and it need to be rewritten, e.g.: the numbering of figures is incorrect (on pages 5-7 the authors presented Figures 1-3, but on page 8 - Figure 1, what is contradictory with reference in the text (in line 289, page 7 is mentioned Figure 4B….???)
The titles of Figures like as the descriptions of results, e.g. page 5, lines: 202-203 the Authors have written: “Expressions of vasoconstrictor enzymes COX1/2, CYP4A and TP receptor are significantly lower in BA, while expression of TXAS is similar BA and MCA arteries”, but in lines 198-200 was mentioned: We have found a significantly lower expression of COX1 and COX2 in BA compared to MCA arteries (Fig1A and B)”. Similarly, in lines 240-241 are mentioned the same information as in lines 233-235. Similar comments refer to Figure 3 on page 7.
The numbering of tables (page 8-9) is incorrect. Additionally, in these tables, the Authors compared their results with literature data, which should not have happened in the Results section, but in Discussion. The comments dealing with the literature data (lines 316-317: Literature data indicated that these 9 genes encode mediators of WSS-induced vasoconstriction or vasodilation (Table 2)) should be moved to Discussion section.
Due to the above, the presentation of results is difficult to follow for it.”
Response 2
Thank you for your insightful comment that “the Authors compared their results with literature data, which should not have happened in the Results section, but in Discussion”. We acknowledge that our present data were not directly compared with literature data. However, we have added supplementary information to our findings that support the role of the genes studied in the present study. We believe that incorporating published data into our findings, as part of data integration and comparative analysis, enhances the strength of our conclusions by allowing for a direct comparison, thereby strengthening the relevance and reliability of our findings. Nevertheless, we agree with the reviewer that these sentences belong to the Discussion, thus we moved accordingly (Page 18, lines 611-616).
In addition, as you suggested, we checked all Figures and Tables for correctness and corrected the typos and mistakes in our revised manuscript.
Reviewer 3 Report
Comments and Suggestions for Authors
Authors present an animal study on 45 Wistar rats which were decapitated and their basilar artery (BA) and cerebri media artery (MCA) were isolated for analyis of concentration of mediators of flow by transcriptome analyis of ExCA and InCA from rats (n=25) assessed by RNA sequencing. In BA compared to MCA, COX1/2, and Cyp450 protein expressions
were lower, PGIS was higher, TXAS and nNOS/eNOS were similar, TP receptor was lower, IP receptor was higher. Gene expressions of vasodilator pathways were higher in extracranial circulation, i.e. in basilar artery. For evaluation of Materials and Methods a specialist in animal studies, i.e. molecular biologist is needed. Introduction provides sufficient information. Results are blurred by the simple fact that BA is not extracranial vessel in humans, and the hypothesis of the authors is that it has more place to dilate than MCA; authors should thoroughly explain and correct this - if BA is extracranial in rats, that is fine, but would it not be more suitable to use intracranial and extracranial ICA? clinical application of these findings is completely missing
Author Response
Dear Prof. Dr. Smania,
Thank you for serving as the editor on our manuscript (ID: life-3571395) entitled "Differential gene and protein expressions responsible for vasomotor signaling provide mechanistic bases for the opposite flow-induced responses of extra - and intracranial arteries". We are grateful for the helpful comments and suggestions of the Reviewers. Each comment and our response are provided in the following section. All modifications are indicated in red text in the revised manuscript.
Authors Introduction to Reviewers Comments:
The authors would like to thank the Reviewers for their thoughtful and insightful comments. We addressed each comment point by point in the following section. In response to these comments, we have amended our revised manuscript. We feel that these modifications have substantially improved the revised manuscript.
Responses to Reviewer 3 comments:
“Authors present an animal study on 45 Wistar rats which were decapitated and their basilar artery (BA) and cerebri media artery (MCA) were isolated for analyis of concentration of mediators of flow by transcriptome analyis of ExCA and InCA from rats (n=25) assessed by RNA sequencing. In BA compared to MCA, COX1/2, and Cyp450 protein expressions were lower, PGIS was higher, TXAS and nNOS/eNOS were similar, TP receptor was lower, IP receptor was higher. Gene expressions of vasodilator pathways were higher in extracranial circulation, i.e. in basilar artery.”
Comments 1: For evaluation of Materials and Methods a specialist in animal studies, i.e. molecular biologist is needed.
Response 1: We agree with the reviewer. In response to it we have got evaluated the Materials and Methods with a molecular biologist. All corrections are indicated in the manuscript with red fonts.
Comments 2: Introduction provides sufficient information.
Response 2: We thank the reviewer for this recognition.
Comments 3: Results are blurred by the simple fact that BA is not extracranial vessel in humans, and the hypothesis of the authors is that it has more place to dilate than MCA; authors should thoroughly explain and correct this - if BA is extracranial in rats, that is fine, but would it not be more suitable to use intracranial and extracranial ICA?
Response 3: The Reviewer’s point is well taken. We have amended our manuscript by replacing the terms “intracranial (InCA)”and “extracranial (ExCA) cerebral arteries” with “cerebral arteries of the posterior cerebral circulation (cerebral arteries of the PCC or CAPCC)”, and “cerebral arteries of the anterior cerebral circulation (cerebral arteries of the ACC or CAACC)” throughout the manuscript in red fonts.
In the study we determined the protein expression of main AA derived vasoactive enzymes and receptors in the basilar artery to compare with the middle cerebral arteries. For transcriptomic gene expression analysis, we collected one cerebral artery of the arteries supplying the circle of Willis, such as the basilar artery, to compare with the arteries posterior circle of Willis, such as the middle cerebral arteries (originating from the internal carotid arteries). The vessels were systematically collected at a consistent time each day to minimize possible circadian rhythm influences on expression patterns. Based on these, we have revised our manuscript using the terms PCC, which includes the posterior cerebral arteries, the posterior communicating arteries, the vertebral arteries and the basilar artery; and ACC, which includes the internal carotid arteries, the anterior cerebral arteries, the anterior communicating arteries, and the middle cerebral arteries.
We have revised the title of the manuscript with the following title
“Differential gene and protein expressions provide mechanistic bases for the opposite responses of pre - and post - circle of Willis arteries to hemodynamic forces”
We have inserted the following sentences in “Abstract” paragraph (page 1, lines …): “Increases in flow elicit dilations in the basilar artery (BA) supplied by the posterior cerebral circulation (PCC), and ensuring efficient blood supply to the circle of Willis in which blood flow and pressure can distribute and equalize thus provide appropriate supply for daughter branches reaching certain brain areas. In contrast, increases in flow elicit constrictions in the middle cerebral artery (MCA), supplied by the anterior cerebral circulation (ACC) and regulating the blood pressure and flow in distal cerebral circulation.”; (page 1, lines 24-29): “
In “Introduction” (page 2, lines 50-61): “…vasomotor functions of the posterior cerebral circulation (PCC), such as the basilar artery (BA) and in the anterior cerebral circulation (ACC), such as the middle cerebral artery (MCA). Cerebral arteries of the PCC (CAPCC) system provide increased or decreased blood flow to the circle of Willis, in which the pressure and flow equilibrate ensuring constant supply of cerebral blood flow to distal branches. In contrast, the role of cerebral arteries of the ACC (CAACC) is to prevent great increases in intracranial pressure and flow, as the closed cranium limits changes in blood flow/volume – a principle recognized early on by Monro and Kellie (PMID: 11425944). The constancy of total intracranial volume is – in part – achieved by the autoregulation of cerebral blood flow via pressure - and flow-induced vasomotor function (dilation/constriction), which prevents the transmission of high systemic pressures and flow into the cerebral capillary network (PMID: 12750541, PMID: 21610722, PMID: 34205652, PMID: 32696219, PMID: 35831968).”
We have inserted the following sentences in “Discussion” (page 13, lines 417-419): “… pressure-induced vasoconstriction [25] - by the flow-sensitive function of the cerebral arteries of the anterior cerebral circulation, such as the MCA and penetrating arterioles…”
We have inserted the following sentences in “Discussion” (page 13, lines 411-413) “… cerebral arteries of the posterior circulation (such as the BA) are not affected by these limitations, allowing them to dilate”;
We have removed the following sentence in “Discussion”: “…likely because MCA is located inside the rigid cranium limiting volume changes …”.
We have inserted the following sentence in “Introduction” (page 2, lines 81-83): “…flow-dependent responses are different and opposite in cerebral arteries of the posterior, such as the BA and the anterior cerebral circulation, such as the MCA. “
Comments 4: “clinical application of these findings is completely missing”
Response 4: Thank you for pointing this out. In response to it we have inserted the following sentence with appropriate references in “Discussion” (page18, line 618): “Physiological importance and possible clinical applications of the findings”.
We have inserted the following sentences in “Discussion” in red fonts and a new figure as Figure 5 with figure legends (page 19, lines 641-662):
“Arteries of the anterior cerebral circulation are susceptible to atherosclerosis, which can cause flow limiting stenosis and embolization of plaque distally towards the brain. Embolization of atherosclerotic plaque within the internal carotid artery predominantly obstructs the middle cerebral artery, leading to ischemic stroke. In such cases, compensatory basilar artery dilation could facilitate increased blood flow towards the middle cerebral artery via collateral pathways in the circle of Willis, thereby mitigating the effects of arterial blockage. Redistributing collateral blood flow via vasodilation in the BA (to the posterior cerebral circulation) to enhance collateral circulation through the circle of Willis and reducing vasoconstriction in the MCA could compensate for the reduction of blood flow in the MCA during acute ischemic events (Fig 5).”
We have inserted the following sentences with appropriate references in “Discussion” in red fonts (Pages 19-20, lines 666-682): “There are several possible therapeutic, diagnostic, and medication possibilities of our findings. Administration of pharmacological vasodilators based on prostaglandins (e.g., prostaglandin analogs, such as epoprostenol) could enhance vasodilator capacity of the BA. Inhibition of vasoconstrictor eicosanoids, such as the TXA2, by administration of thromboxane receptor antagonists could attenuate vasoconstriction during ischemic periods. Determination of AA-derived metabolites in the cerebrospinal fluid or blood plasma as biomarkers for imbalanced vasoconstriction and vasodilation could be a diagnostic tool. AA-derived drugs, such as prostacyclin analogs (e.g., iloprost and treprostinil) to enhance vasodilation in the BA, or TXA2 receptor antagonists (e.g., seratrodast) to attenuate vasoconstriction in the MCA, or using selective modulators of prostaglandin receptors (e.g., EP4 agonists) for vascular relaxation, and COX inhibitors to reduce vasoconstrictor metabolites while preserving vasodilator prostaglandins can be used in the medication of stroke patients.
Another possibility is endarterectomy of stenosis in the carotid artery, which likely involves the capacity of the basilar artery to dilate in response to increasing flow, enhancing the inter-hemispheric perfusion through the circle of Willis thereby improving the cerebral circulation.”
Round 2
Reviewer 1 Report
Comments and Suggestions for Authors
This acquisition has very effectively answered my questions. I don't have any other problems at present.
Reviewer 2 Report
Comments and Suggestions for Authors
I accept the manuscript in present form for publication.
Reviewer 3 Report
Comments and Suggestions for Authors
Sufficient response